# Closing the Gaps to Understand the Tick Transmission of *Anaplasma marginale* among Giant Anteaters (*Myrmecophaga tridactyla*) in Argentina

**DOI:** 10.3390/pathogens9121033

**Published:** 2020-12-09

**Authors:** Eliana Carolina Guillemi, Mélody Imbert, Sofia de la Fournière, María Marcela Orozco, Jorge Peña Martinez, Ana Carolina Rosas, Valeria Noely Montenegro, Marisa Diana Farber

**Affiliations:** 1Instituto de Agrobiotecnología y Biología Molecular (IABIMO, INTA-CONICET), Las Cabañas y Los Reseros s/n, Hurlingham, Buenos Aires B1686LQF, Argentina; delafourniere.sofia@inta.gob.ar (S.d.l.F.); montenegro.valeria@inta.gob.ar (V.N.M.); farber.marisa@inta.gob.ar (M.D.F.); 2Toulouse III, Université des Sciences Paul Sabatier, 31330 Toulouse, France; melody.imbert@live.fr; 3Instituto de Ecología, Genética y Evolución de Buenos Aires (IEGEBA-CONICET), Facultad de Ciencias Exactas y Naturales, Universidad de Buenos Aires, Intendente Güiraldes 2160–Ciudad Universitaria Ciudad Autónoma de Buenos Aires, Buenos Aires C1428EGA, Argentina; marcelaorozco.vet@gmail.com; 4The Conservation Land Trust, Corrientes 3470, Argentina; jorgepmartinez@gmail.com (J.P.M.); anacarolinarosas.vet@gmail.com (A.C.R.)

**Keywords:** *Anaplasma marginale*, giant anteater, *Myrmecophaga tridactyla*, *Amblyomma dubitatum*, *A. sculptum*

## Abstract

*Anaplasma marginale*, a well-known cattle pathogen of tropical and subtropical world regions, has been previously molecularly characterized in a giant anteater (*Myrmecophaga tridactyla*) from Corrientes, Argentina. Ticks or other hematophagous arthropod involved in the wild transmission cycle remained unknown. The aim of the present study was to analyze the simultaneous occurrence of *A. marginale* in blood samples and ticks from giant anteaters from Corrientes in order to investigate if ticks could be relevant in the transmission among these mammals. Blood samples from 50 giant anteaters collected in different years and 26 ticks *Amblyomma dubitatum* and *A. sculptum* were studied through the molecular amplification of two unequivocal species-specific genes from *A. marginale: msp5* and *msp1β*. Twenty five giant anteaters and tick organs (salivary glands, gut and oviduct) from 11 ticks tested positive to the *A. marginale* DNA amplification. The further molecular characterization through MSP1a tandem repeats analysis revealed the presence of genotypes circulating among giant anteaters that had been previously identified in cattle blood samples from the same geographical region. These results confirm the presence of *A. marginale* in giant anteaters in Corrientes and suggests that *A. dubitatum* and *A. sculptum* ticks could be involved in the transmission among giant anteaters. Future studies will determine the role of these tick species in the wild transmission cycle in the study area and the eventual connection with the domestic cycle.

## 1. Introduction

Anaplasmosis, caused by the bacterium *Anaplasma marginale*, is an endemic disease of cattle in tropical and subtropical regions, that can be either biologically transmitted by ticks or mechanically spread by blood contaminated fomites [1]. *Anaplasma marginale* (alpha Proteobacteria, Rickettsiales, *Anaplasmataceae*) is an obligate intraerythrocytic pathogen that causes moderate to severe hemolytic anemia, leading to significant economic losses to beef and dairy industry due to lower weight gain rate, lower milk production, abortions, high treatment costs and mortality [1]. Particularly, in Argentina, anaplasmosis is widespread north of latitude or parallel 33°S and is mainly transmitted by *Rhipicephalus microplus* ticks [2].

Though other domestic and wild ruminants than bovine had been reported to be infected by *A. marginale* [3,4,5,6,7,8] non ruminant host species could be playing a significant role in the epidemiology of the disease. In a previous study, our group reported the detection of *A. marginale* DNA in a blood sample from a giant anteater (*Myrmecophaga tridactyla*) and the identification of spherical intraerythrocyte inclusions suggestive of *A. marginale* in blood smears. The strain was characterized through MSP1a tandem repeats analysis and a multi locus sequence typing (MLST) scheme [9]. Furthermore, Calchi et al. [10], reported the molecular identification of *Anaplasma* spp. in Xenarthra mammals (including *M. tridactyla*) from Brazil.

The giant anteater *Myrmecophaga tridactyla* (Xenarthra, *Myrmecophagidae*) is the species with most records (*n* = 5941) among anteaters in the Neotropics and, like other Xenarthrans (sloths, armadillos and other anteaters), has essential functions for ecosystem maintenance, such as insect control and nutrient cycling, playing key roles as ecosystem engineers [11]. In Argentina, *M. tridactyla* is distributed in the north of the country [11] and currently there is a reintroduction program that aims to re populate a native territory in Corrientes province.

The aim of the present study was to analyze the presence of *A. marginale* not only in a greater number of anteater blood samples, but also in on-host ticks, in order to investigate the transmission cycle of *A. marginale* among giant anteaters.

## 2. Results

### 2.1. Detection of A. marginale DNA from Blood Samples and Ticks

From the 50 blood samples studied, 25 tested positive for two *A. marginale* specific genes (Figure 1). The anteaters positive for *A. marginale* have been sampled in four different locations and throughout the different years studied (Figure 1 and Appendix A).

We collected 26 ticks from eight giant anteaters (Table 1) together with a host blood sample (except for GA 42). We identified two tick species: *A. dubitatum* (*n* = 12) and *A. sculptum* (*n* = 14), and we selected only the adult stage (*n* = 23) to perform dissection. The 23 adult ticks were identified as females (seven *A. dubitatum* and 11 *A. sculptum*) and males (5 *A. dubitatum*). We considered *A. marginale*-infected ticks the ones that tested positive for both PCRs targeting *A. marginale* specific fragments in at least one organ (six *A. dubitatum* and five *A. sculptum*) (Table 2). These 11 positive ticks corresponded to 6 giant anteaters, four of which also tested positive for *A. marginale* in the blood sample (Table 1).

### 2.2. Molecular Characterization of A. marginale Strains

The molecular amplification and sequencing of a fragment of the *msp1α* gene, was used for the characterization of 11 *A. marginale* strains (Table 3). We identified two different haplotypes based on 5 repeat profiles (Table 4). The nucleotide sequences from *msp1α* gene were deposited in GenBank (accession numbers pending).

## 3. Discussion

The comprehensive study of the anaplasmosis epidemiology needs the identification of all or the most significant elements involved, including susceptible, carrier and potential reservoir hosts, as well as all the potential vectors. In the present study we tested for the presence of *A. marginale* in the blood samples and the ticks from giant anteaters from Corrientes province, Argentina. For the samples we tested, we found half of the Xenarthra mammal bloods (25 out of 50) to be positive for *A. marginale*. All the collected ticks corresponded to the genus *Amblyomma* and the two species identified, *A. dubitatum* and *A. sculptum*, had been previously reported parasitizing *M. tridactyla* [12]. Half of the dissected adult *A. dubitatum* and *A. sculptum* ticks tested positive for the bacteria through two PCR assays targeting two different *A. marginale* specific gene fragments. Though most of the positive ticks were collected from positive mammals hosts we got two unmatched samples. That is, for the anteater GA42 we had no blood sample, however we detected an *A. marginale* positive tick. For the anteater GA22, the tick samples linked to this animal were positive for *A. marginale* in different tissues (including the SG), but the blood tested negative. Particularly, in this last case we hypothesized that this scenario could correspond to an infected questing tick collected on host before attaching, taking into account the three-host cycle of *A. dubitatum* [12].

Ticks from the genus *Amblyomma* were also collected from *M. tridactyla* in the study reported by Calchi et.al. from Brazil [10]. *Amblyomma nodosum* ticks were found in the *Anaplasma* sp. positive giant anteaters, even though these arthropods were not studied for the presence of *Anaplasma* sp.

The presence of *A. marginale* DNA in *A. dubitatum* and *A. sculptum* organs (salivary glands, gut and oviduct) suggests that, after being acquired in a blood meal, this bacterium could replicate and spread through the tick tissues, especially in the salivary glands from where it could be transmitted to another susceptible host. The three-host parasitic cycle of *Amblyomma* [12] supports the possibility that *A. dubitatum* and *A. sculptum* could act as vectors in the wild transmission cycle of *A. marginale* in Corrientes province. Moreover, as *A. dubitatum* and *A. sculptum* also parasite cattle, both tick species could be responsible for the connection between the domestic and the wild transmission cycle of *A. marginale*.

Anaplasmosis is an endemic disease that affects livestock in Argentina and previous studies have been carried out for the molecular characterization of *A. marginale* genotypes from cattle located in some of the areas where the studied giant anteaters are found [2,13,14]. In the present study, molecular characterization based on MSP1a tandem repeats analysis revealed the presence of two already known haplotypes. Thus, using *msp1α A. marginale* specific molecular marker we are not only unequivocally confirming the detection of *A. marginale*, but also suggesting the potential link of domestic and the wild cycle, as long as the same kind of genotypes have been previously reported in *A. marginale* strains from cattle [2,15]. Further studies are needed to better characterize the *A. marginale* isolates from *M. tridactyla* to fully understand the epidemiology and both the domestic and the wild cycle of transmission.

Regarding the genus *Anaplasma* in the wildlife, there are reports of *A. marginale* and other species in the region. In a recent surveillance study of *Blastocerus dichotomus* (marsh deer) morbidity and mortality in Argentina, *A. marginale, A. platys, A. odocoilei*, and Candidatus *Anaplasma boleense* were found [16]. These findings are in accordance with previous reports for the same deer species and *Mazama gouazoubira* from Brazil, in which *A. marginale* and *Anaplasma spp.* were reported [6]. Agents from de genus *Anaplasma spp* were also identified in *Nasua nasua* (coati), domestic dogs, *Cerdocyon thous* (crab-eating fox), *Leopardus pardalis* (ocelot), wild rodents, and marsupials from Brazil [17]. Interestingly, *Anaplasma* spp. has been also reported in *Tayassu pecari* (peccaries) and *Bradypus tridactylus* (sloths) from Brazil [18], being the last one, another species from the super order Xenarthra.

*Anaplasma marginale* has been previously reported in other *Amblyomma* ticks species as *A. variegatum*, *A. cajennense* and *A. maculatum* [19,20], but to the best of our knowledge, this is the first report of *A. marginale* infecting *A. dubitatum* and *A. sculptum*. Moreover, previous reports of *A. marginale* in *Amblyomma* ticks have been performed using DNA extracted from the whole ticks, not from dissected organs, this kind of approach may not be the most suitable for determining the tick infection since the amplified *A. marginale* DNA could come from the residual blood meal.

Although *M. tridactyla* is listed as vulnerable by the International Union for the Conservation of Nature (IUCN) [21] and by the Red List of mammals of Argentina [22], the species was extinct in the Corrientes province. The “Giant Anteater Reintroduction Project” started in 2007 and aims to repopulate a native territory for *M. tridactyla*. Even though it is known that for all xenarthran species the major threats are habitat loss resulting from fragmentation, domestic and feral dog attacks, roadkill, subsistence hunting, illegal capture and fires [11], the role of unstudied pathogens and the spillover effect from domestic animals to wildlife should be studied, especially in this recently reintroduced giant anteater subpopulation.

Whether *A. marginale* represents a threat to the giant anteater’s health remains to be studied but, all the results in the present study suggest that *A. dubitatum* and *A. sculptum* ticks could be involved in a wild cycle of transmission of *A. marginale* among giant anteaters as hosts and also that these tick species could be implicated as a link with the domestic cycle of cattle.

## 4. Materials and Methods

### 4.1. Giant Anteaters Blood Samples and Ticks

Blood samples (*n* = 50) and ticks (*n* = 26) from giant anteaters were received and analyzed in our laboratory for *A. marginale* identification. Samples corresponded to giant anteaters sampled between 2007 and 2018 at different time points and at four different locations in the province (Appendix A) in the framework of the “Giant Anteater Reintroduction Project” (http://www.proyectoibera.org/especiesamenazadas_osohormiguero.htm). The four locations were: El Socorro (28°39′09,2″S–57°25′48,5″W), San Cayetano (27°33′06″S–58°40′40,8″W), San Alonso (28°18′22,5″S–57°27′30,1″W) and Don Pablo (29°31′30,7″S–59°01′01,8″W). Procedures for animal management have been conducted under Argentinian and provincial protocols (TP Nº1140/15) in the framework of the “Recovery Plan for the Giant Anteater in the Esteros de Iberá” and in accordance with the Guidelines of the American Society of Mammalogists for the use of wild mammals in research [23,24]

### 4.2. Ticks Taxonomic Identification

The ticks were identified using taxonomic keys [12] under a stereoscopic magnifier (10X–40X). After identification, only adult ticks were dissected in order to recover salivary glands (SG), gut (GUT) and the oviduct (OV) in case of female ticks. For this purpose, ticks were first washed in sterile PBS solution and then dissected under the stereoscopic magnifier using sterile instrumental. Organs were individually washed in sterile PBS solution and then conserved separately at 4 °C in labeled tubes containing PBS sterile solution until DNA extraction.

### 4.3. DNA Extraction from Blood Samples and Ticks

DNA from giant anteater blood samples and adult tick organs was extracted by phenol/chloroform method followed by a standard ethanol precipitation. Briefly, samples (400 µL of whole blood and the whole tick organ) were incubated with a volume (400 µL for blood samples and 200 µL for each organ) of an extraction buffer (Tris-CLH: 100 mM, SDS: 1%, NaCl: 300 mM, EDTA: 25 mM, pH: 7.5) at 37 °C for one hour, then 200 ng/µL Proteinase K (INBIO HIGHWAY. Tandil, Argentina) was added to the mix and incubated over night at 56 °C to finally proceed with the phenol/chloroform extraction [25]. DNA quality and concentration were determined using a microvolume spectrophotometer (NanoDrop ND-1000. ThermoFisher Scientific. Waltham, MA, USA). DNA from nymph ticks was not extracted since correct dissection was not possible.

### 4.4. Detection of A. marginale DNA from Blood Samples and Ticks

For *A. marginale* identification, two unequivocal specie-specific genes were targeted: *msp5*, a single copy gene that encodes the outer major surface protein MSP5 and *msp1β* a three-copy gene that encodes the outer major surface protein MSP1b from *A. marginale*. For both target genes, PCR reactions were conducted using primers previously reported [26,27]. The molecular amplifications were performed in a 50 µL reaction mixture (0.4 µmol of each primer, 0.2 mM of each deoxyribonucleotide triphosphate, 1.25 U of TopTaq DNA polymerase (QIAGEN. Hilden, Germany), 5 µL of 10× PCR buffer and purified water for 50 µL of final volume) using 200 ng of genomic DNA (from both blood or tick samples). Amplifications were carried out in a thermocycler (Bio-Rad MyCycler Thermal Cycler. Hercules, CA, USA) under previously described cycling conditions. For each amplification reaction, positive (DNA from *A. marginale* Mercedes strain) and negative (pure water) controls were included. An aliquot of 5 µL of each amplified product was analyzed by electrophoresis in 1.5% agarose gel stained with ethidium bromide. A molecular size marker (1 Kb Plus DNA Ladder, Invitrogen. Carlsbad, CA, USA) was used to determine PCR product size.

### 4.5. Molecular Characterization of A. marginale

To further characterize the isolates, a fragment of the *msp1α* gene was amplified and sequenced. The *msp1α* gene encodes the outer major surface protein MSP1a from *A. marginale,* and the number and type of tandem repeats in the 5′region of this gene defines the *A. marginale* genotypes [28]. The PCR reactions and the amplicons visualization were performed as described in the Section 4.4. For those positive samples, the remaining 45 µL of the amplification product were purified using a commercial kit (QIAquick. PCR Purification Kit. QIAGEN).

Both strands from each amplicon were sequenced with a Big Dye Terminator v3.1 kit from Applied Biosystems (Foster City, CA, USA) and analyzed on an ABI 3130XL genetic analyzer from the same supplier (Genomic Unit, Consorcio Argentino de Tecnología Genómica (CATG), Instituto de Biotecnología, CICVyA, INTA). Raw files from the target gene (forward and reverse chromatogram in ab1 format) were processed using the Vector NTI Advanced 10 program (Invitrogen). Both chromatograms were used for assembling a contig. The final file in FASTA format was used for further sequence analysis including translation into an amino acid sequence that enabled the repeats profile identification.

## 5. Conclusions

In the present study, *A. marginale* has been undoubtedly identified in blood samples from 25 giant anteaters from Corrientes, Argentina, and in the tissues from *A. dubitatum* and *A. sculptum* ticks collected from those mammals. Together with previous evidence, we confirm the presence of *A. marginale* infecting this non ruminant host species, suggesting a wild alternative host in the transmission cycle of this bacterium, in which *A. dubitatum* and *A. sculptum* ticks could be implicated.

## Figures and Tables

**Figure 1 pathogens-09-01033-f001:**
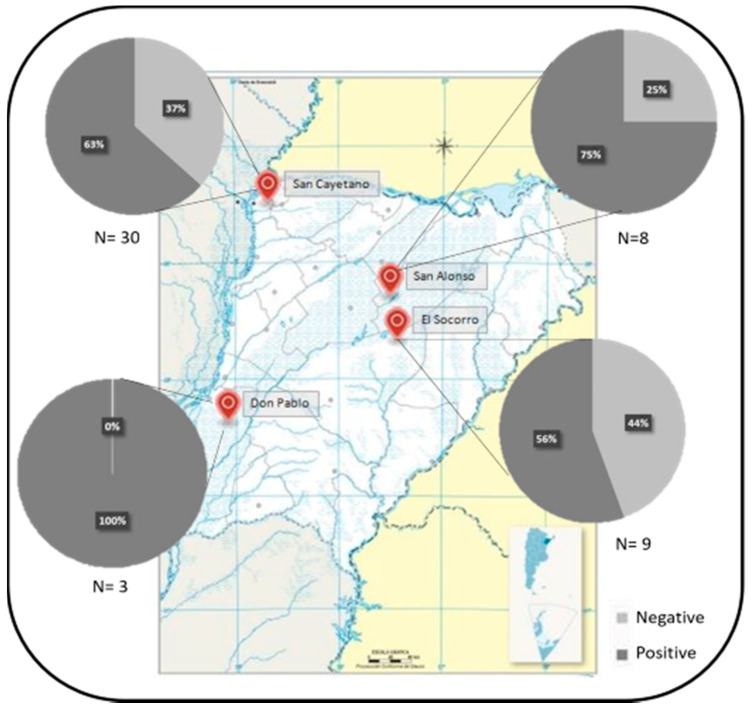
Number, origin and results of *A. marginale* (positive/negative) for the giant anteater blood samples. In each location the total blood samples analyzed is indicated (*n*) and each graph shows the percentage of positive (dark gray) and negative (light gray) samples.

**Table 1 pathogens-09-01033-t001:** Identification of tick species found parasitizing giant anteaters and discrimination of their different life stages. The results for *A. marginale* molecular identification from giant anteater blood samples are given in the second column.

Tick Species		*Amblyomma dubitatum*			*Amblyomma sculptum*		
Giant Anteater identification number	*A. marginale* in the blood sample	F	M	N	F	M	N
GA16	Positive	1					
GA22	Negative	4	2				
GA31	Positive	1					
GA33	Positive						3
GA35	Positive		2		3		
GA36	Positive				6		
GA37	Positive		1		2		
GA42	No sample	1					
Total		12			14		

GA: giant anteater; F: female; M: male, N: nymph.

**Table 2 pathogens-09-01033-t002:** Results for *A. marginale* molecular identification from tick tissues.

Giant Anteater Identification Number	Tick Species	Stage	Tick Organ		
			SG	OV	GUT
GA16	*A. dubitatum*	F	Am		
GA22	*A. dubitatum*	M		/	Am
	*A. dubitatum*	M	Am	/	
GA35	*A. dubitatum*	M	Am	/	
GA36	*A. sculptum*	F	Am		
	*A. sculptum*	F	Am	Am	Am
	*A. sculptum*	F			Am
	*A. sculptum*	F			Am
	*A. sculptum*	F	Am	Am	
GA37	*A. dubitatum*	M	Am	/	Am
GA42	*A. dubitatum*	F	Am		

GA: giant anteater; SG: salivary glands; OV: oviduct; GUT: gut; Am: *A. marginale* detection.

**Table 3 pathogens-09-01033-t003:** *Anaplasma marginale msp1α* genotypes from 11 strains identified in giant anteaters blood samples.

Giant Anteater Identification Number	Genotype
GA 5	13 27
GA 6	13 27
GA 10	13 27
GA 12	13 27
GA 13	13 27
GA 14	13 27
GA 16	13 27
GA 18	23 30 31 31 31
GA 19	13 27
GA 21	13 27
GA 24	13 27

**Table 4 pathogens-09-01033-t004:** Amino acid sequence for MSP1a tandem repeats identified in *A. marginale* strains from giant anteaters.

Repeat	Encoded Sequence
13	TDSSSASGQQQESSVLSQSDQASTSSQLG
23	TDSSSASGQQQKSSVLSQSSQASTSSQLG
27	ADSSSASGQQQESSVLSQSDQASTSSQLG
30	ADSSSASGQQQKSSVLSQSSQASTSSQLG
31	ADSSSAGNQQQESSVSSQSDASTSSQLG

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
