# Peer review of "Closing the Gaps to Understand the Tick Transmission of Anaplasma marginale among Giant Anteaters (Myrmecophaga tridactyla) in Argentina"

_pathogens, 2020, doi:10.3390/pathogens9121033_

Round 1

Reviewer 1 Report

This is a concise study to detect Anaplasma marginale in the giant anteater and ticks from a region in Argentina.

The study design is good, methods used are appropriate with proper controls and the presentation of results is good. The discussion section is also precise. However, I would like to see other examples of the wild life cycle of Anaplasma spp.

There are frequent editorial and English language errors which need to be addressed.

The references MUST be formatted as per the journal's style. 

Author Response

We are grateful to the Editor and the Reviewers for their attentive reading of our manuscript and helpful insight. Please find below the detailed responses to the reviewers’ comments and find the corrections in the revised version of the manuscript.

Reviewer 1

This is a concise study to detect Anaplasma marginale in the giant anteater and ticks from a region in Argentina.

The study design is good, methods used are appropriate with proper controls and the presentation of results is good. The discussion section is also precise. However, I would like to see other examples of the wild life cycle of Anaplasma spp.

Thank you for your comments. We have added more information about Anaplasma spp. in wildlife from the region in the discussion section. Specifically, we added a report with the identification of Anaplasma spp. in another Xenarthra species, the sloth Bradypus tridactylus from Brazil (Soares 2017).

There are frequent editorial and English language errors which need to be addressed.

We carefully revised the document to correct these errors.

The references MUST be formatted as per the journal's style. 

References are now edited based on the journal's style

Reviewer 2 Report

Reviewer comments for Guillemi et al., 2020. Closing the gaps to understand the tick transmission of Anaplasma marginale among giant anteaters (Myrmecophaga tridactyla) in Argentina

Overall reviewer comments:
It a pleasure to have been requested to review this paper by Guillemi et al., 2020, which implicates two Amblyomma tick species, A. dubitatum and A. sculptum in the transmission of A. marginale, a globally important tick-borne pathogen of ruminants mostly, in the giant anteater (Myrmecophaga tridactyla). The manuscript endeavours to demonstrate the presence of A. marginale in DNA extracted from tissues dissected the aforementioned tick species as well as their anteater host species, using A. marginale-specific PCR amplification of the two A. marginale genes, msp1-beta and msp5. It is my considered view that while the authors present interesting and worthwhile research that sheds light on and give important information about previously unknown tick players in the transmission of A. marginale in giant anteaters, the claims they make that they have detected A. marginale DNA in both the tick-tissues and their giant anteater hosts, need more information to undoubtedly prove this fact. The reviewer believes that the absence of sequence information arising from the PCRs they carried out, which is clarified in the comment I make below, is a major shortcoming of this manusript as this results in a lack of adequate information to fully support their findings.

It is the reviewer's view that the outcomes of this research would have benefited greatly from sequencing of the msp1-beta and msp5 gene PCR products/ amplicons generated, followed by a BLAST analysis in order to fully confirm the identity of the sequences generated. This information is lacking in the manuscript and needs to be added to it to remove any doubt in their the findings. These sequences may have further have been utilised in a phylogenetic analysis, in the event that they showed high levels of difference in sequence identity with previously deposited A. marginale sequences in the GenBank database. These approaches could have also given important information about how these giant anteater sequences compare with or relate to other A. marginale sequences.

Further comments/ queries on the methodology of the manuscript:
Were ticks allowed to digest their blood meal prior to dissection for DNA extraction, in order to show vector competence or a role of A. dubitatum and A. sculptum in biological transmission of A. marginale? Could this approach have been useful to clarify the A. marginale infection status of the nymph stages of the ticks you found? Also, did you wash the tick tissues, prior to DNA extraction? In my view these measures may help to elucidate the role of the Amblyomma tick species as biological vectors or transmission agents of A. marginale, while ruling out the possibility of contamination from host blood in the tick. Also, given the difficulty of morphological identification of ticks, amplification and sequencing of genes such as the mitochondrial 16S rRNA, could have also helped in the identification of the tick species you found on the anteater hosts.

Comments and suggested changes on other text:

Line 26: Change to '...species-specific genes...'

Line 42: Change to 'Particularly, in Argentina, anaplasmosis is widespread...' no need to for ',' after anaplasmosis

Line 42: Change to '...north of latitude or parallel...'

Line 45: Change to '...could be playing a significant role...'

Line 55: Change to ' In Argentina, M. tridactyla is...'

Line 59: Change to '...number of anteater blood samples...'

Line 70: Change to '...together with a host blood sample...'

Line 71-72: Change to 'We identified two tick species...and we selected only the adult stage.' This improves the flow of the paragraph.

Line 73-74: Change to 'We considered A. marginale-infected ticks the ones...'

Lines: 77-79: Change Table 1 title to: 'Identification of tick species found parasitising giant anteaters and discrimination of their different life stages. The results for A. marginale molecular identification from giant anteater blood samples are given in the second column'

Table 1, column 1 heading: Change to 'Giant Anteater Identification number'. This also applies to Table 2, column 1.

Line 81: Change to '...from tick tissues.'

Line 84: Change to '...identification of all or the most significant elements involved...'

Line 87-88: Comment on your statement, 'Half of the Xenarthra mammal’s population (25 out of 50) was positive for A. marginale.' - It seems to the reviewer that you tested a sample or subset of the full population of Xenarthra in this study area and not the whole population. If this is correct, then the above statement may be misleading. I would, therefore, suggest changing this statement to say 'For the samples we tested, we found half of the Xenarthra mammal bloods (25 out of 50) to be positive for A. marginale.'

Line 94: Replace 'anyhow', with 'however'

Line 94: Change '...the linked ticks...' to '...the tick samples linked to this animal...'

Line 95: Change to '...were positive for A. marginale...'

Line 105: Change to '...supports the possibility...'

Line 110: Change to '...livestock in Argentina and previous studies have been carried out for the molecular...'

Line 121: Change to '...listed as vulnerable by the...'

Line 122: Change to '...the species was extinct in the Corrientes province.'

Line 123: Change 're populate' to one word 'repopulate'

Line 124: 'Even though...'

Line 131: The use of the word 'holded' makes the meaning of this sentence unclear. Are you implicating A. dubitatum and A. sculptor as biological vectors responsible for the transmission of A. marginale in giant anteaters here? Please clarify.

Author Response

We are grateful to the Editor and the Reviewers for their attentive reading of our manuscript and helpful insight. Please find below the detailed responses to the reviewers’ comments and find the corrections in the revised version of the manuscript.

Reviewer 2

Reviewer comments for Guillemi et al., 2020. Closing the gaps to understand the tick transmission of Anaplasma marginale among giant anteaters (Myrmecophaga tridactyla) in Argentina

Overall reviewer comments:
It a pleasure to have been requested to review this paper by Guillemi et al., 2020, which implicates two Amblyomma tick species, A. dubitatum and A. sculptum in the transmission of A. marginale, a globally important tick-borne pathogen of ruminants mostly, in the giant anteater (Myrmecophaga tridactyla). The manuscript endeavours to demonstrate the presence of A. marginale in DNA extracted from tissues dissected the aforementioned tick species as well as their anteater host species, using A. marginale-specific PCR amplification of the two A. marginale genes, msp1-beta and msp5. It is my considered view that while the authors present interesting and worthwhile research that sheds light on and give important information about previously unknown tick players in the transmission of A. marginale in giant anteaters, the claims they make that they have detected A. marginale DNA in both the tick-tissues and their giant anteater hosts, need more information to undoubtedly prove this fact. The reviewer believes that the absence of sequence information arising from the PCRs they carried out, which is clarified in the comment I make below, is a major shortcoming of this manusript as this results in a lack of adequate information to fully support their findings.

This is a very interesting comment that no doubt deserves consideration. Since the study of Anaplasmosis in giant anteaters needs to further being investigated, we are still working with the samples and trying to answer new questions. Unfortunately, at current we are working in the genotypic characterization of the strains but we still do not have the complete information. This stage seems to be more difficult due to methodological difficulties inherent to the molecular markers, but, in addition, the COVID-19 pandemics situation is restricting us to go further in this characterization. However, we understand that adding this partial information (for 11 samples) will help to support our results for Anaplasma marginale in giant anteaters samples. For this reason, we added all the information required (materials and methods, results tables and discussion) regarding this additional information. Accession numbers are pending, we have already submitted the sequences to GenBank but we are waiting for the accession numbers, we will include them as son as we get them.

It is the reviewer's view that the outcomes of this research would have benefited greatly from sequencing of the msp1-beta and msp5 gene PCR products/ amplicons generated, followed by a BLAST analysis in order to fully confirm the identity of the sequences generated. This information is lacking in the manuscript and needs to be added to it to remove any doubt in their the findings. These sequences may have further have been utilised in a phylogenetic analysis, in the event that they showed high levels of difference in sequence identity with previously deposited A. marginale sequences in the GenBank database. These approaches could have also given important information about how these giant anteater sequences compare with or relate to other A. marginale sequences.

The additional target gene that we amplified and sequenced corresponds to the msp1α molecular marker that encodes the Major Surface Protein MSP1a. This marker defines the genotypes based on different type and number of tandem repeats, so that the data that we get are not able to be used in phylogenetic analysis. Instead, we provieded the nucleotide sequences and the amino acid repeats.

Further comments/ queries on the methodology of the manuscript:
Were ticks allowed to digest their blood meal prior to dissection for DNA extraction, in order to show vector competence or a role of A. dubitatum and A. sculptum in biological transmission of A. marginale? Could this approach have been useful to clarify the A. marginale infection status of the nymph stages of the ticks you found?

We apologize if this part of the text was confusing. We didn´t allowed ticks to digest their blood meal, once sampled they were conserved in tubes containing 70° alcohol and sent to our laboratory. We only dissect adult specimens. The problems associated to nymphs dissection are due to their size and the difficulty to correctly remove each tissue separately. The fact that we found A. marginale DNA in the tick’s salivary glands and oviduct let us hypothesize the role of A. dubitatum and A. sculptum in A. marginale transmission.

Also, did you wash the tick tissues, prior to DNA extraction? In my view these measures may help to elucidate the role of the Amblyomma tick species as biological vectors or transmission agents of A. marginale, while ruling out the possibility of contamination from host blood in the tick.

Thank you for noticing this omission. We first washed the entire tick in sterile PBS and then each organ was individually washed in new sterile PBS solution. We added this information in section 4.2 of materials and methods.

Also, given the difficulty of morphological identification of ticks, amplification and sequencing of genes such as the mitochondrial 16S rRNA, could have also helped in the identification of the tick species you found on the anteater hosts.

Thank you for this helpful comment. The identification of tick species in our region is well described based on taxonomic keys that have been recently updated (Nava et al 2017). On the other hand, both identified tick species (A. sculptum and A. dubitatum) have been previously reported in M. tridactyla, which represents an important epidemiological support. Although molecular evidence could confirm our taxonomic tick identification, we consider it unnecessary in this context.

Comments and suggested changes on other text:

Line 26: Change to '...species-specific genes...'  Correction made.

Line 42: Change to 'Particularly, in Argentina, anaplasmosis is widespread...' no need to for ',' after anaplasmosis Correction made.

Line 42: Change to '...north of latitude or parallel...' Correction made.

Line 45: Change to '...could be playing a significant role...' Correction made.

Line 55: Change to ' In Argentina, M. tridactyla is...' Correction made.

Line 59: Change to '...number of anteater blood samples...' Correction made.

Line 70: Change to '...together with a host blood sample...' Correction made.

Line 71-72: Change to 'We identified two tick species...and we selected only the adult stage.' This improves the flow of the paragraph. Correction made.

Line 73-74: Change to 'We considered A. marginale-infected ticks the ones...' Correction made.

Lines: 77-79: Change Table 1 title to: 'Identification of tick species found parasitising giant anteaters and discrimination of their different life stages. The results for A. marginale molecular identification from giant anteater blood samples are given in the second column' Correction made.

Table 1, column 1 heading: Change to 'Giant Anteater Identification number'. This also applies to Table 2, column 1. Correction made.

Line 81: Change to '...from tick tissues.' Correction made.

Line 84: Change to '...identification of all or the most significant elements involved...' Correction made.

Line 87-88: Comment on your statement, 'Half of the Xenarthra mammal’s population (25 out of 50) was positive for A. marginale.' - It seems to the reviewer that you tested a sample or subset of the full population of Xenarthra in this study area and not the whole population. If this is correct, then the above statement may be misleading. I would, therefore, suggest changing this statement to say 'For the samples we tested, we found half of the Xenarthra mammal bloods (25 out of 50) to be positive for A. marginale.' Correction made.

Line 94: Replace 'anyhow', with 'however' Correction made.

Line 94: Change '...the linked ticks...' to '...the tick samples linked to this animal...' Correction made.

Line 95: Change to '...were positive for A. marginale...' Correction made.

Line 105: Change to '...supports the possibility...' Correction made.

Line 110: Change to '...livestock in Argentina and previous studies have been carried out for the molecular...' Correction made.

Line 121: Change to '...listed as vulnerable by the...' Correction made.

Line 122: Change to '...the species was extinct in the Corrientes province.' Correction made.

Line 123: Change 're populate' to one word 'repopulate' Correction made.

Line 124: 'Even though...' Correction made.

Line 131: The use of the word 'holded' makes the meaning of this sentence unclear. Are you implicating A. dubitatum and A. sculptor as biological vectors responsible for the transmission of A. marginale in giant anteaters here? Please clarify.

Based on the findings of A. marginale DNA in the tick’s tissues, we hipothesize that both Amblyomma tick species are involved in the transmisión cycle of this bacterium at least among giant anteaters. However, more studies are needed to confirm this claim. As a consequence, we preferred to attenuate our statement by saying that: ´ Whether A. marginale represents a threat to the giant anteater’s health remains to be studied but, all the results in the present study suggest that A. dubitatum and A. sculptum ticks could be involved in a wild cycle of transmission of A. marginale among giant anteaters as hosts´